# Exploring effects of response biases in affect induction procedures

**David J. Moulds, Jona Meyer, Janet F. McLean, Vera Kempe** *

Division of Psychology, School of Applied Sciences, Abertay University, Dundee, United Kingdom

* v.kempe@abertay.ac.uk

## Abstract

This study examined whether self-reports or ratings of experienced affect, often used as manipulation checks on the efficacy of affect induction procedures (AIPs), reflect genuine changes in affective states rather than response biases arising from demand characteristics or social desirability effects. In a between-participants design, participants were exposed to positive, negative and neutral images with valence-congruent music or sound to induce happy, sad and neutral mood. Half of the participants had to actively appraise each image whereas the other half viewed images passively. We hypothesised that if ratings of affective valence are subject to response biases then they should reflect the target mood in the same way for active appraisal and passive exposure as participants encountered the same affective stimuli in both conditions. We also tested whether the AIP resulted in mood-congruent changes in facial expressions analysed by FaceReader to see whether behavioural indicators corroborate the self-reports. The results showed that while participants' ratings reflected the induced target valence, the difference between positive and negative AIP was significantly attenuated in the active appraisal condition, suggesting that self-reports of mood experienced after the AIP are not entirely a reflection of response biases. However, there were no effects of the AIP on FaceReader valence scores, in line with theories questioning the existence of cross-culturally and inter-individually universal behavioural indicators of affective states. Efficacy of AIPs is therefore best checked using self-reports.

## Introduction

The experimental study of how affective states like emotions and moods affect behaviour relies on affect induction procedures (AIPs) to elicit specific affective states. AIPs expose participants to affectively charged stimuli, sometimes combined with instructions to try to 'feel' the targeted affective states. Their efficacy is typically checked through some form of self-reports, often using various rating scales. However, such self-reports have been found to be subject to demand characteristics [1–5] or social desirability effects [6, 7]: when exposed to affectively charged stimuli participants may align their self-reports with expectations on how to act in accordance with, or sometimes even in opposition to, what they perceive a study's hypothesis to be or how to react in a socially desirable manner given the stimuli they have just been exposed to. Thus, researchers cannot be sure whether participants really experience the desired affective state or whether they merely purport to do so. A meta-analysis of 250 effects of

**Data Availability Statement:** The data underlying the results presented in the study are available from https://osf.io/fqu7p/.

**Funding:** This study was supported by an RLINCS-studentship from Abertay University to the first

author. The funders had no role in study design, data collection and analysis, decision to publish, or preparation of the manuscript.

**Competing interests:** The authors have declared that no competing interests exist.

induced positive and negative affective states found that outcomes were weaker when demand characteristics were controlled either by deceiving participants about the true purpose of the experiment, by including demand control groups where participants were asked to behave *as if* they were experiencing a certain mood or by telling them to expect the opposite effect [5]. However, hiding the true purpose of an AIP may not be sufficient to eliminate response biases as the nature of the stimuli may still cause participants to respond in line with normative social expectations.

Note that in this paper we are mainly concerned with affective valence and will therefore use the more general term 'affective states'. We use the term 'mood' when referring to the transient mild changes in affective states induced by induction procedures in the laboratory, and the term 'emotion' whenever referring to theoretical accounts that prefer this term in order to emphasise that many affective states are triggered by events in the environment.

Discussions of whether demand characteristics and social desirability effects may limit the validity of AIPs were more prominent at the end of the last century, especially in conjunction with the Velten [8] technique [9, 10] in which participants read self-referential statements to induce happiness or sadness. More recently, these discussions seem to have faded from attention even though the issues are far from settled, and most manipulation checks of AIPs still use self-reports. Self-reports, especially rating scales, are particularly prone to demand effects [2, 3, 5], especially when participants are told of the aim of the procedure or receive instructions on which affective states they are meant to experience. Thus, effects of AIPs appear stronger when their efficacy is measured by self-reports rather than by behavioural or physiological measures [11, 12]. Here we investigated whether self-reports elicited after AIPs are subject to such response biases by presenting the same AIP in two conditions–passive exposure vs. active appraisal of the affective stimuli–to see whether it changed self-reported mood. This approach was chosen because a control of demand effects by trying to deceive participants is often difficult to implement in a believable fashion. We reasoned that differences in self-reports depending on appraisal condition must reflect differences in genuinely experienced mood because participants were exposed to the same affective stimuli in both conditions but have no expectations about potential effects of the appraisal condition. If, however, self-reports mainly reflect response bias, then we would expect no effect of the appraisal manipulation as the affective stimuli are identical.

We also explored whether behavioural indicators such as facial expressions confirm the results of the self-reports. If so, we would expect that facial expressions also reflect the target mood differently in the two appraisal conditions. Facial expressions can be measured either using intrusive procedures like facial electromyography or, less intrusively, using standardised coding of facial actions, e.g., by employing the Facial Action Coding System (FACS [13, 14]). However, human coding of facial expressions using FACS is costly as it requires extensive certified training of the coders [13, 14]. Recently, automated forms of facial expression analysis, such as Noldus' FaceReader software (version 6.1 [15]) have come to greater prominence as they eliminate the need for extensive training of human experts. Compared to AIP manipulation checks using physiological measures, such as facial electromyography, skin conductance or heart rate (e.g. [16]), automatic scanning of facial expressions by FaceReader has the benefit of being unobtrusive as it requires only a webcam. FaceReader is intended to work on a variety of hardware options (PC, laptop) and is marketed as a versatile instrument for emotion identification. Unobtrusiveness and versatility are especially useful for AIPs that involve deceiving participants into believing that a study has a different goal from affect induction, which should reduce effects of demand characteristics on the behaviour under scrutiny (e.g. [17, 18]). For these reasons, we used facial expressions identified by FaceReader as a supposedly more objective indicator of AIP efficacy.

FaceReader tracks changing affective states reflected in facial expressions in real-time, with approximately 30 observations per second of video input. It computes scores for intensity of movement in facial action units for the facial expressions of a neutral state and the basic emotions of happiness, sadness, anger, surprise, fear and disgust. Valence scores are computed as the intensity difference between positive and negative emotions, which can range from -1 to 1. Arousal scores are computed as the overall intensity of facial action unit activation, which can range from 0 to 1. FaceReader was trained with manually annotated facial images [19] and has been tested and validated with images from the Radboud Faces Database (RaFD [20]). Further validation studies [21, 22] showed an emotion recognition accuracy between 79% and 88%, slightly above that of human coders, for still images of posed facial emotion expressions. These expressions were enacted either without training [22], using the Stanislavski-method [23] or by deliberately engaging pre-specified facial action units [24]. Accuracy was particularly high for expressions of happiness.

Evidence that FaceReader can detect affective states induced by AIPs is less consistent. Noordewier et al. [25] showed a reliable increase in FaceReader valence scores using a within-participants repetition-change paradigm where a succession of neutral visual stimuli was suddenly interrupted by a surprising stimulus (e.g., a picture of a cute puppy or a wounded vs. a funny face). These effects were mainly due to a subset of individuals with particularly high facial expressivity. Höfling et al. [26] compared FaceReader scores with measures of affective valence based on fEMG and with measures of arousal based on skin conductance during an AIP that exposed the same participants to pleasant, unpleasant and neutral images from the International Affective Picture System [27]. Their study found that FaceReader was as reliable as fEMG in detecting positive affect, although it showed the valence effect only after about 2 seconds while fEMG showed an effect right after image presentation. Although FaceReader valence scores were positively correlated with fEMG and subjective valence ratings of the images, they did not differ between negative and neutral images, suggesting that FaceReader has reduced sensitivity to negative affect, even within participants. Furthermore, FaceReader arousal scores were positively correlated with skin conductance measures and with subjective arousal ratings of the images, but showed a significant increase only for negative images while skin conductance showed an increase for both positive and negative compared to neutral images. These findings suggest that FaceReader can detect induced positive affect and low arousal with a reasonable degree of sensitivity but performs worse in detecting induced negative affect and high arousal.

Such inconsistency is not surprising given that there are theoretical and empirical reasons to doubt the existence of a reliable link between subjective feelings and behavioural or physiological indicators. From a theoretical point of view, the notion that affective states manifest themselves in objectively measurable behaviours and physiological states is predicated on essentialist views of emotions as states of an organism that share a causal mechanism, e.g., as an evolved adaptation to specific environmental phenomena that trigger universally recognisable bodily and mental responses [28–32]. An example is the facial expressions of the six "basic emotions" of happiness, sadness, surprise, fear, anger, and disgust [33]. Yet researchers find it difficult to agree on what these basic emotions are (e.g. [34]) and what their observable manifestations across a range of physiological and behavioural indicators should be [35]. The weak link between affective states and behaviours can be demonstrated empirically, for example, for facial expressions in naturalistic settings [36], where there are only weak correlations between emotions and predicted facial expressions (e.g. [37–39]).

Reasons to doubt that facial expressions are reliable indicators of emotions are provided by the Theory of Constructed Emotion [40], which proposes that emotions are associated with sets of predictions of sensations that arise from interoceptive and exteroceptive inputs. These

predictions, performed by multiple neural networks [41, 42], may fit specific situations based on past experiences, and form categories that serve to construct meanings that allow the sensations to be perceived subjectively as specific emotions. Such meaning-creating categories can be socially shared and labelled, just like colour words are socially shared categories that carve up the range of sensations evoked by the colour spectrum. Thus, there is diversity in the associations between patterns of neural activity, conceived of as statistical summaries of a particular sample from a broader population of possible neural activation patterns, and specific affective states [43]. While associations of patterns of neural activation with specific emotion categories have been demonstrated (e.g. [44, 45]), these associations often fail to replicate across different studies [42].

The Theory of Constructed Emotion has implications for measurement–construction of emotions as categorisations over variable sensory experiences makes consistent identification of these categories from behavioural manifestations or from physiological responses impossible, especially when these 'objective' markers (e.g., facial expressions or skin conductance) are measured in isolation. In short, the relationship between feelings and physiology is, at best, ambiguous [46] and, at worst, a conceptual mistake [35]. The same emotional categories can be experienced subjectively while individuals display a wide variety of different physiological responses [40, p. 14–15; 47], and similar physiological or behavioural responses such as facial expressions can be experienced as very different emotional states [40, p. 12; 48 49–50]. Thus, under this view, lack of a correlation between self-reports and facial expressions may merely attest to the futility of identifying emotions from objective 'fingerprints' such as facial expressions. This implies that self-reports should be given primacy as a means to ascertain whether an individual's subjective experience corresponds to the culturally accepted emotion expected to be elicited by a specific situation. It is therefore even more important to understand the extent to which self-reports provide unbiased accounts of an individual's subjective experience of their affective state.

## The present study

We presented participants with affective images accompanied by music to induce positive, neutral and negative affective states. This type of AIP had previously been shown to elicit a significant difference in self-reports between positive and negative states [51, 52]. We compared self-reports of affective valence between conditions where participants were asked to actively appraise the valence of each image (active appraisal) and a condition where they simply viewed the images (passive exposure). We also included self-reports and FaceReader evaluations of arousal, irrespective of valence, to make sure that the targeted valence manipulations were not confounded by differences in arousal.

If the affective appraisal of stimuli itself contributes to the experience of affective states [50, 53], we expect the experienced affective valence to differ between the Active Appraisal and Passive Exposure conditions. Consequently, a significant difference in self-reports between appraisal conditions would suggest that the specific features of an AIP—e.g., whether it raises participant awareness of the targeted affective state via continuous appraisal—influences participants' subjective experience. It would mean that the AIP achieved, at least in part, a genuine alteration of affective state and that self-reports can be informative about relative group differences in affective state elicited by AIPs in between-participant designs. Note that this conclusion would hold regardless of the direction of the difference between appraisal conditions: For example, active appraisal could attenuate reported mood by increasing cognitive load [54], thereby hindering the processing of affective stimuli. It is also possible that participants in the passive exposure conditions exaggerate their reported affect as they may construe mood

questionnaires after the AIP as inquiring about their appraisal of the affective valence of the images, rather than their own feelings, while participants in the active appraisal conditions, who will have already performed such ratings, can focus their responses on their felt experience, thereby showing genuine AIP effects over and above any possible demand characteristics. Crucially, if participants' self-reports reflect genuinely experienced affective states rather than demand characteristics or social desirability, the one outcome we would not expect is for the continuous appraisal of the images to make no difference at all (cf. [53]); rather, any difference between appraisal conditions would point to a genuine component in the reporting of induced affective states. This should manifest itself in a significant interaction between induced mood (Happy, Neutral, Sad) and appraisal condition (Active Appraisal vs. Passive Exposure). Moreover, if behavioural responses, such as facial expressions, are more reliable indicators of experienced affective states than self-reports, as suggested by essentialist theories of emotion, the FaceReader valence scores should reflect the target affective valence and should interact with the appraisal conditions.

If, on the other hand, reports of experienced mood after an AIP represent effects of demand characteristics or social desirability, rather than a genuine reflection of subjective affective state, then asking participants to appraise the valence of stimuli should not make a difference to their self-reports and there should be no interaction between induced affective states and the appraisal conditions: Similar self-reports regardless of appraisal condition would indicate that what participants report after the AIP may be based on the perceived valence of the images rather than the valence of their genuinely felt mood. If FaceReader valence scores are genuine indicators of experienced affect, they also should not differ if the AIP fails to induce genuine affect.

Finally, under the Theory of Constructed Emotion, a third outcome is possible—that AIPs induce genuine affect but facial expressions are not genuine indicators of such altered subjective states. In this case, we would expect an interaction between the induced affective states and the appraisal conditions in the self-reports, but no effects of either manipulation on the FaceReader valence scores.

## Method

### Participants

One hundred and twenty-two participants were recruited; 62 indicated their gender as male and 60 as female. Participants were recruited and participated between July 2016 and March 2020, alongside other studies taking place in our lab. Ten male and ten female participants were assigned to each of six conditions. However, due to a coding error, one extra male participant was recruited and assigned to one condition (Happy mood, Active appraisal). One remaining male participant was excluded for not following the task instructions. This left ten male and ten female participants each in five conditions, and eleven males and ten females in the sixth condition. The 61 male participants ranged in age from 18 to 60 (M = 25.79, SD = 7.42), and the 60 female participants ranged in age from 18 to 59 (M = 26.77, SD = 9.66). Except for matching for gender, participants were randomly assigned to the six conditions created by the crossing of Mood (Sad vs. Neutral vs. Happy) and Appraisal (Passive Exposure vs. Active Appraisal). All participants were native speakers of English, or were staff or students of UK universities studying and/or teaching in English and were therefore judged to have a high degree of proficiency in the English language. Due to the Covid-19 pandemic and resulting lockdown measures in the UK towards the end of data collection, one participant completed the experiment remotely. This participant connected to the experimenter's laptop using Team-Viewer (version 15.3.8497 [55]) in order to access the AIP. This made it impossible to record

the participant on video, but they completed the self-report questionnaire and their valence and arousal ratings were not outliers.

The study received clearance by the Ethics Committee at Abertay University. Twenty one participants refused consent for being video-recorded and the one remote participant could not be video-recorded; these 22 were excluded from the FaceReader analysis. Due to the video recordings, researchers had access to personally identifiable information (facial recordings) for the 99 participants who were included in the FaceReader analysis. Full written consent was obtained, and recordings were stored securely and treated with strict confidentiality.

## Materials and measures

**Affect induction.**   Eighty-four black-and-white images were gathered from publicly available internet sources and rated for affective valence by another seven male (mean age = 25.57, SD = 5.44 years) and seven female (mean age = 29.71, SD = 6.60 years) in a pilot study that took place in 2012/ 2013. These participants did not participate in the main experiment. They were asked to rate each image on a scale from 1 (very sad) to 7 (very happy). From these pre-rated images, the eight highest rated images (i.e. the most positive ones), eight lowest rated (i.e. the most negative ones), and eight rated closest to four (i.e. the most neutral ones) were selected and included in the AIP.

All images were gathered from the Internet in 2012 and included scenes containing people. The happy images depicted people who were visibly friends or lovers exhibiting positive affect, or cute babies, alone or interacting with their mother. The sad images depicted victims of war, famine, and natural disasters. The neutral images depicted people in urban or professional contexts, with no overt emotional expressions visible. The images were viewed on desktop or laptop computer monitors and presented using E-prime (version 2.0 [56]). In the Happy and Sad mood conditions, the eight mood-congruent images were presented with the eight neutral filler images included to provide some variability in affective content for the rating task. Each image was presented twice alongside a piece of mood-congruent music. In the Happy mood conditions, participants listened to Mozart's Serenade Nr. 13 for Strings in G "Eine kleine Nachtmusik" [57]. In the Sad mood conditions, participants listened to John Williams' "Schindler's list: Theme" [58]. In the Neutral mood conditions, the eight neutral images were presented four times instead of twice. Instead of mood-congruent music, participants listened to crowd noise taken from "Big crowd chatter" [59]. The order of images was randomised in all six AIP conditions. The accompanying sound began as soon as the first image appeared, and continued on repeat until the end of the AIP. Each image occupied 80% of the screen area, and was presented fully centred against a black background.

**Manipulation checks.**   Upon completion of the AIP, valence and arousal were assessed using a self-report scale based on the Brief Mood Introspection Scale (BMIS [60]). The original BMIS consists of eight negative (*fed up*, *nervous*, *sad*, *jittery*, *drowsy*, *grouchy*, *tired and gloomy*) and eight positive (*happy*, *lively*, *calm*, *active*, *content*, *caring*, *loving and peppy*) mood adjectives. Following Kempe et al. [61] the word "peppy" was replaced with "bubbly", because "peppy" was found to be unfamiliar to contemporary British participants, based on feedback from pilot studies. The 16 adjectives are rated on a four-point scale including "definitely do not feel", "do not feel", "slightly feel", and "definitely feel", which is not very sensitive to the subtle variation in affective states. For the present study, we therefore amended the BMIS to be measured on a visual analogue scale by creating the Brief Mood Introspection Scale—Visual Analogue (BMIS-VA). To increase the scope for variability and aggregation and to ensure an even number of positive and negative valence, low and high arousal adjectives, we amended the BMIS-VA by removing four of the original BMIS items (*fed up*, *grouchy*, *caring*, *and loving*) and adding eight new items in order to produce four balanced subscales. The BMIS-VA

consists of five high arousal, positive valence adjectives (*excited*, *lively*, *active*, *joyful*, and *bubbly*); five low arousal, positive valence adjectives (*happy*, *serene*, *calm*, *content*, and *relaxed)*; five high arousal, negative valence adjectives (*nervous*, *jittery*, *annoyed*, *fearful*, and *angry*); and five low arousal, negative valence adjectives (*bored*, *sad*, *drowsy*, *gloomy*, and *tired*). The visual analogue scale was a single horizontal line printed across the page, with "definitely do not feel" printed to the left, and "definitely feel" printed to the right. The adjective for the participant to rate was printed centrally above the corresponding visual analogue scale. Participant responses were measured to the nearest half millimetre, from 0 to 116mm (inclusive), meaning that this scale was essentially a 233-point scale instead of the original 4-point version. The BMIS-VA was presented on one two-sided piece of paper, with an example line (pre-marked in the exact neutral position) and the first nine adjectives on one side, and the remaining eleven adjectives presented on the other side. The 20 items of the BMIS-VA were presented in one of three pre-prepared randomised orders. On occasions where participants did not know one or more of the adjectives, these were defined to them according to the free Oxford Dictionary of British and World English [62]. Minor amendments to the layout of the BMIS-VA questionnaire that became necessary over the course of the experiment are described in S1 Appendix and the final version is provided at https://osf.io/fqu7p/.

While completing the AIP, participants' faces were recorded. Due to circumstances beyond the control of the experimenters the first half of the sample completed the experiment on a desktop PC with a webcam mounted in the top middle of the desktop monitor, and the second half on a laptop with internal camera. Both devices were placed at the same distance from the edge of the desk in front of the participant. The video was saved and imported into FaceReader [15] for analysis. Participants who were uncomfortable with being video-recorded during the experiment were able to opt out of this part while still completing the rest of the experiment.

## Procedure

Participants were told that they would be taking part in a pilot study to find out how individuals perceive the emotional content of images. Participants were then asked to give consent to viewing the images and to being video-recorded (as detailed above, not all participants consented to video-recording). After receiving instructions pertaining to their Appraisal condition, participants put on headphones and clicked anywhere on the screen to start the presentation of the images. If participants agreed to be video-recorded, the recording was started immediately prior to the AIP; if not, they proceeded without video-recording.

In the Passive Exposure conditions, participants were told they would see a series of images on the computer, and that "for a future study, we need to find out how happy or sad these images make people feel". After each image had been presented for three seconds, an asterisk appeared and participants were instructed to click when they saw the asterisk. Each image then remained on screen for two seconds, before being replaced by a black screen for one second and then the next image.

In the Active Appraisal conditions, participants were instructed to "view each image carefully and try to feel the emotion that the image evokes, then click on the scale". An example of the visual analogue rating scale was then shown, with "very sad" and "very happy" marked at the endpoints, and "neutral" marked in the middle. Following Lang et al. [63], the term "happy" was defined as "happy, pleased, satisfied, contented, hopeful", and the term "sad" was defined as "unhappy, annoyed, unsatisfied, melancholic, despaired". Each image was presented for three seconds before the scale appeared at the top of the screen. After participants clicked on the scale to rate each image, the image remained on screen for a further two seconds before being replaced by a black screen for one second, followed by the next image.

Note that this presentation regime ensured that the exposure time was similar in both conditions and participants had to perform a mouse click after each exposure. During the AIP the experimenter was hidden from the participants' view either behind a screen or in an adjacent room so as not to influence participant mood. Immediately following the AIP, participants completed the BMIS-VA, before being debriefed to explain that the true aim of the experiment was affect induction. Participants who viewed the Sad AIP were offered to view the Happy version in case they still felt sad and wished to put themselves in a better mood before leaving the laboratory (this was not a part of the experiment and no data were collected at this stage).

## Results

Reliability analyses were conducted using SPSS (version 22). All other analyses were conducted in R (version 4.1.1).

### BMIS-VA self-reports

To check the reliability of positive (*happy*, *serene*, *calm*, *content*, *relaxed*, *excited*, *lively*, *active*, *joyful*, and *bubbly*) and negative (*bored*, *sad*, *drowsy*, *gloomy*, *tired*, *nervous*, *jittery*, *annoyed*, *fearful*, and *angry*) valence attributes ratings, Cronbach's alpha was computed for the average of positive and negative valence attribute ratings separately. For the positive attributes, alpha was .92. For the negative attributes, alpha was .74. As both scales scored above .7 they were judged to be reliable. For each participant, an affective valence score was then computed by subtracting the average ratings for the negative attributes from the average of the positive attributes. Higher scores on the BMIS-VA indicated more positive affective valence, with scores above zero representing overall positive affect and below zero overall negative affect. To rule out that the AIP induced changes in arousal rather than in affective valence, we also computed arousal scores by subtracting the average ratings for the low arousal items (*bored*, *sad*, *drowsy*, *gloomy*, *tired*, *happy*, *serene*, *calm*, *content*, and *relaxed*) from the average of the high arousal items (*nervous*, *jittery*, *annoyed*, *fearful*, *angry*, *excited*, *lively*, *active*, *joyful*, and *bubbly*). Unsurprisingly, given the increased salience of valence compared to arousal in the AIP, Cronbach's alpha for both of these scales were low (for high arousal scale, $\alpha$ = .40, for low arousal scale $\alpha$ = .42).

Means and standard deviations of self-reported valence and arousal scores are given in Table 1. Valence and arousal scores were analysed using linear models with treatment coding of Mood and Appraisal and Neutral as well as Happy mood as reference category. We also performed model comparison using ANOVA with Type II sums of squares.

The linear regression model (see S2 Appendix: Table 1) with Neutral mood as reference category showed that during Passive Exposure, self-rated valence was higher than Neutral mood in the Happy mood condition, $\beta$ = 31.38, 95% C.I. = 11.24–51.53, p = .003, and lower than Neutral mood in the Sad mood condition, $\beta$ = -47.48, 95% C.I. = -67.63 –-27.34, p < .001 (see Fig 1). For Neutral mood, the Active Appraisal condition did not differ significantly from

**Table 1. Means and standard deviations (in brackets) of self-reported valence and arousal scores.**

| Self-Report Measure | Appraisal Condition | Mood | | |
|---|---|---|---|---|
| | | Sad | Neutral | Happy |
| Valence | Passive Exposure | -27.86 (25.83) | 19.62 (33.56) | 51.00 (19.36) |
| | Active Appraisal | -6.64 (27.71) | 15.66 (35.53) | 36.68 (45.67) |
| Arousal | Passive Exposure | -8.96 (17.34) | -12.09 (23.42) | -14.95 (14.28) |
| | Active Appraisal | -18.04 (15.67) | -17.12 (12.58) | -10.91 (27.99) |

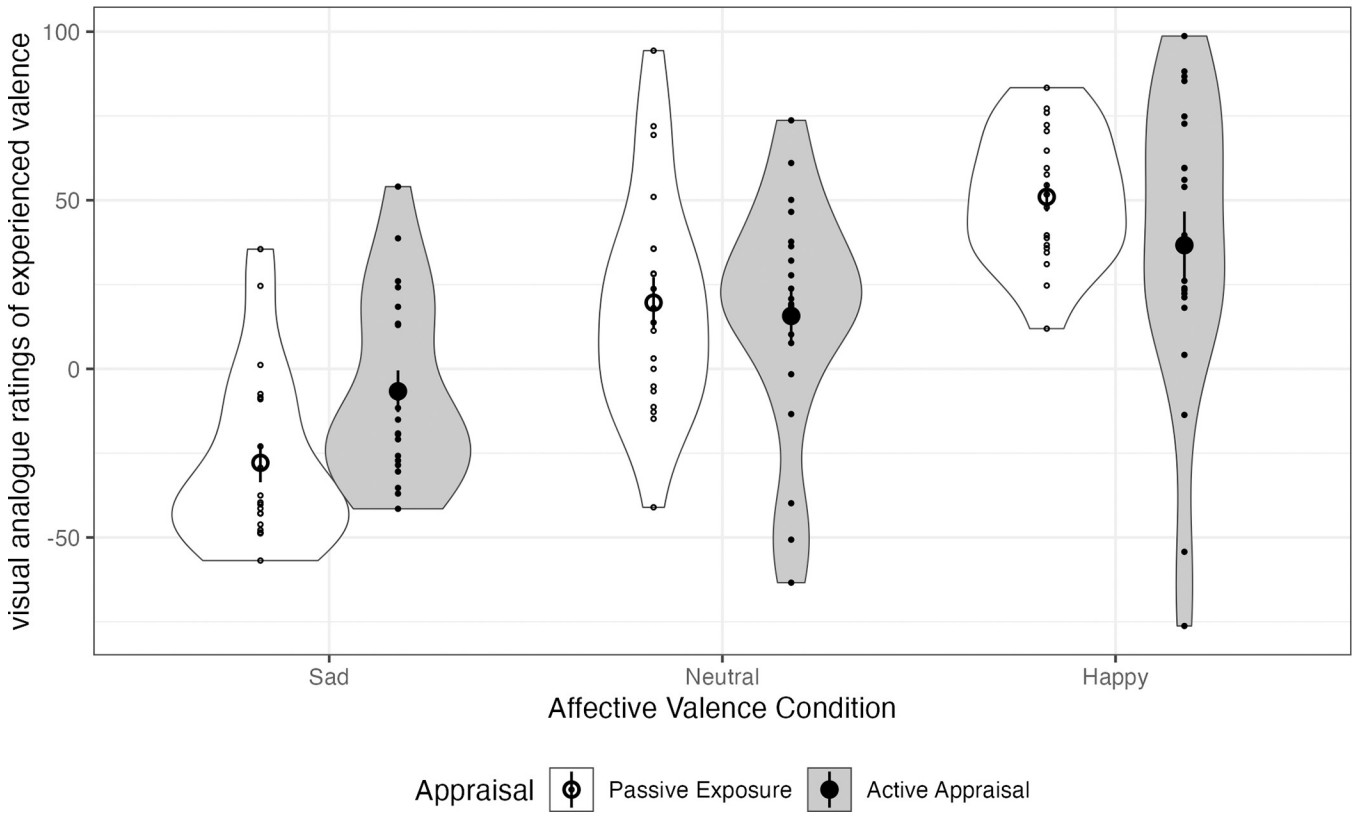

**Fig 1. Distributions of self-rated valence scores.** Means are given by large points; standard errors by lines.

Passive Exposure. The lack of significant interactions suggests that the differences between Happy and Neutral and between. Sad and Neutral were not smaller in the Active Appraisal compared to the Passive Exposure condition. However, when Happy mood was entered as the reference category (S2 Appendix: Table 2), the analysis showed a significant interaction (β = 35.54, 95% C.I. = 7.22–63.86, p = .015) suggesting that the difference in self-rated valence between Happy and Sad mood was significantly smaller in the Active Appraisal compared to the Passive Exposure condition.

Model comparisons using ANOVA confirmed a main effect of Mood, $F(1,115) = 35.8$, $p < .001$, and a significant interaction between Mood and Appraisal. $F(2,115) = 3.19$, $p = .045$. One-way ANOVAs for each Mood condition separately showed significant effects of Mood (for Passive Exposure: $F(2, 57) = 43.6$, $p < .001$; for Active Appraisal: $F(2, 58) = 6.95$, $p = .002$), suggesting that Happy and Sad mood were successfully elicited in either appraisal condition but the difference between these two states was attenuated in the Active Appraisal condition.

For Arousal (S2 Appendix: Table 3), the same linear regression models yielded no significant effects, confirming that the AIP did not induce changes in arousal in any of the appraisal conditions (see Fig 2).

### Facial expression of affect

**Treatment of video data.** *Videos were trimmed using Windows Live Movie Maker to align the* videos with the onset and offset of the AIP and were saved as Movie Maker files and published using Windows Live Movie Maker's recommended settings. For videos recorded on a desktop PC, this was 1440x1080 pixel, "standard" aspect ratio, 24.13 Mbps bit rate. For videos

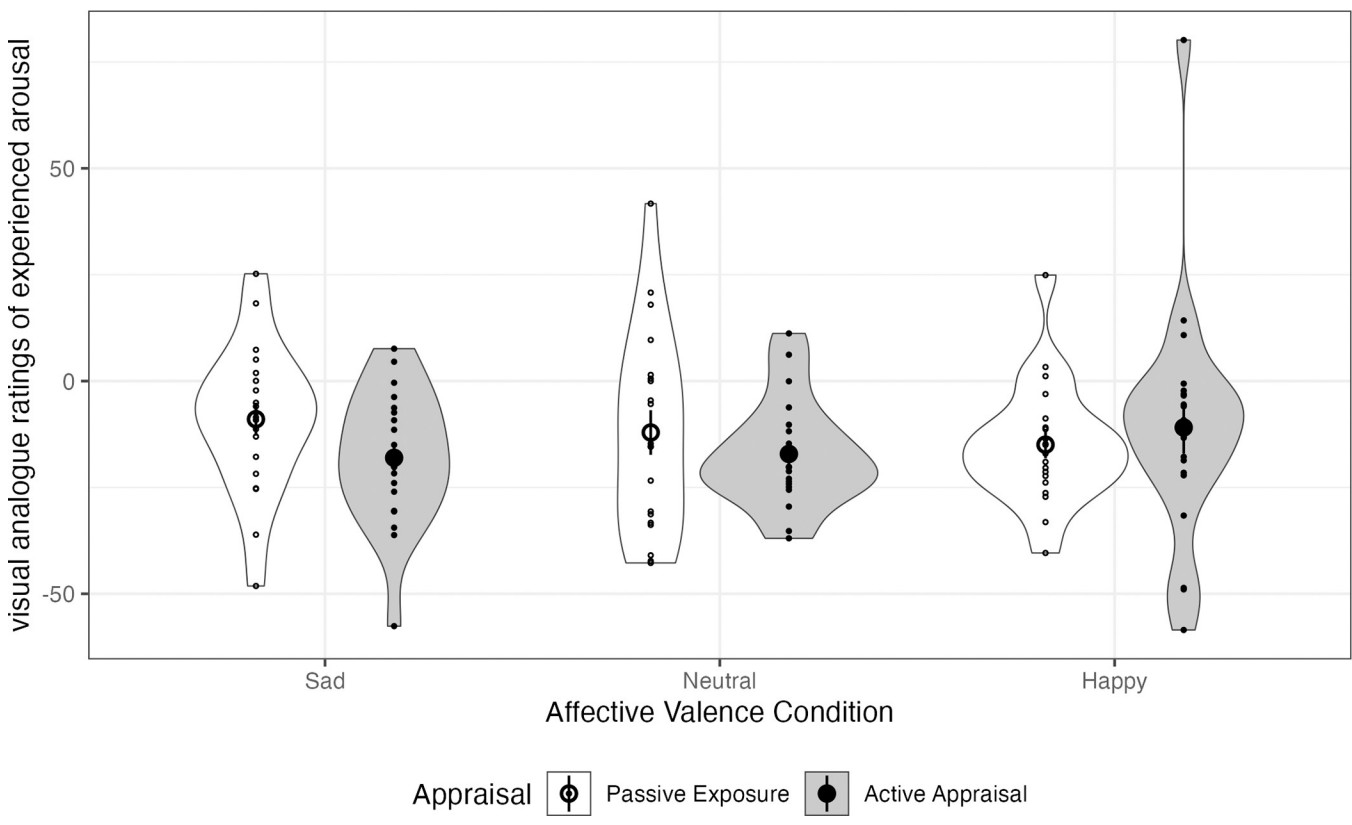

**Fig 2. Distributions of self-rated arousal scores.** Means are given by large points; standard errors by lines.

recorded on a laptop, the quality of the original file was lower, and the recommended settings output from Windows Live Movie Maker were 640x480 pixel, "standard" aspect ratio, 7.13 Mbps. These output settings were selected in order to maintain sufficient quality for FaceReader to detect and analyse facial expressions, while minimising the file sizes for storage. The edited videos were then converted to.mp4 files using Handbrake [64]. This conversion process slightly reduced the bit rate (by varying amounts), but was necessary to enable FaceReader to analyse the videos. However, irrespective of the difference in video quality between those recorded on desktop and laptop computers, and the further reduction in quality caused by the editing and conversion process, FaceReader confirmed the videos were of sufficiently high quality for analysis.

FaceReader analyses video input on a frame-by-frame basis, extracting 30 frames per second. However, frames may not be suitable for analysis if the video quality is poor, or if the subject's face is occluded. Of the 121 participants who took part in the study 21 refused consent to video-recording and the one remote participant did not have their video-data recorded so video data were available for 99. For these 99 participants, the number of valid frames varied considerably, from a minimum of two to a maximum of 13,199, with a mean number of video frames per participant of 6,629.94 ($SD = 2,612.46$). For each suitable frame, FaceReader outputs a score of affective valence and arousal (in addition to classifying the image into the eight emotion categories and generating estimates of how accurately each possible emotional classification would fit). For each participant, we assigned a continuous time stamp to each video frame regardless of suitability so that each suitable frame was analysed as originating from the time in the recording when it actually occurred. This resulted in interrupted time series for

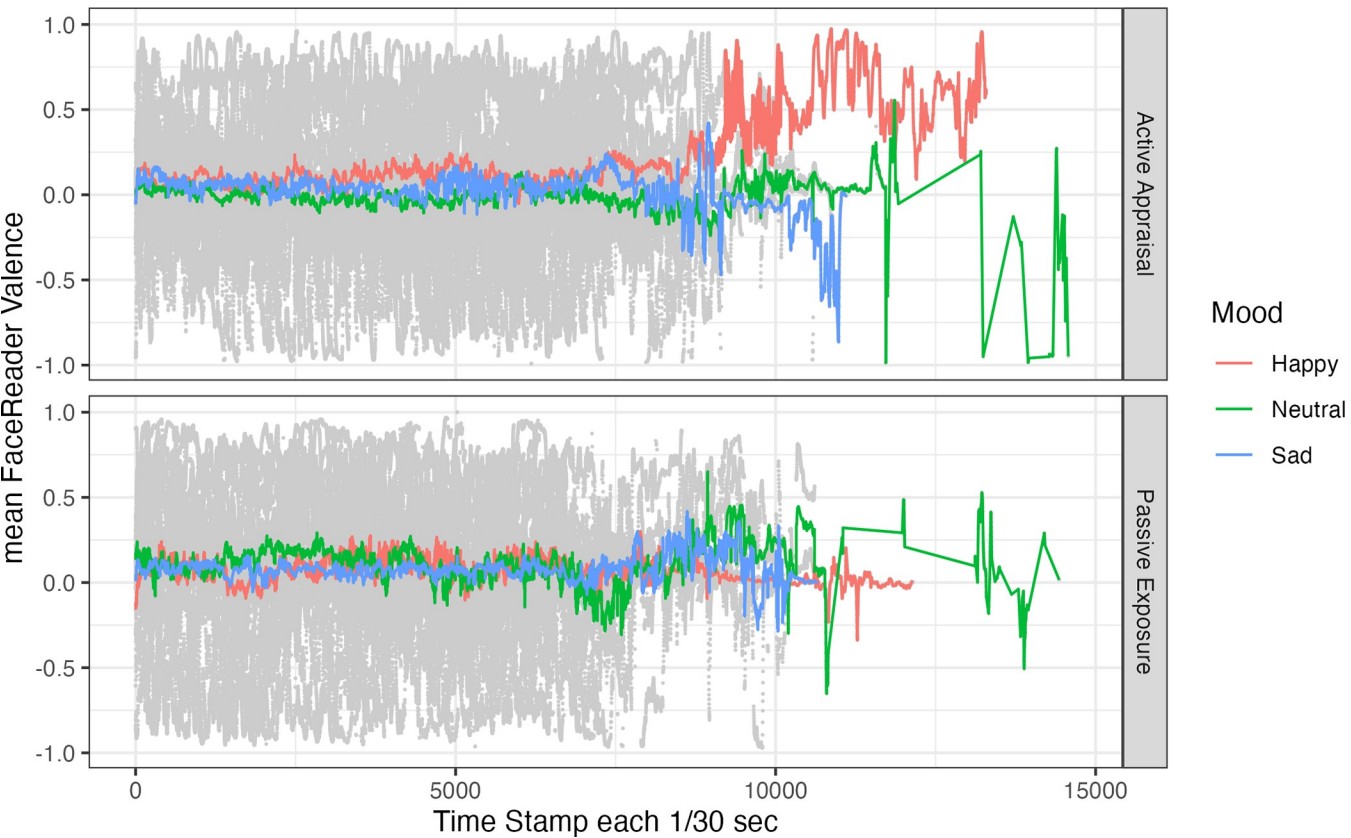

**Fig 3. Mean FaceReader valence scores (in colour) and individual data points by participant (in grey) over time.** *Note.* Sessions differed in length resulting in wider fluctuations of the condition means for later time stamps with fewer data points.

participants with unsuitable frames but preserved the correct time placement of suitable frames resulting in a total of 655,571 frames. All data were subjected to Growth Curve Analyses with Time Stamp as a centred fixed effect to model the change in facial expressions over time. The models also included fixed effects of Mood (reference category: Neutral), and Appraisal (reference category: Passive Exposure), and all interactions as well as random intercepts and slopes of Time Stamp by participant.

**Growth-curve analyses.**   Change of mean valence scores over time as a function of Mood and Appraisal condition are depicted in Fig 3. For valence, the model (S3 Appendix: Table 1) showed no significant effects.

Change of mean arousal scores over time as a function of Mood and Appraisal condition are depicted in Fig 4. For arousal, the model (S3 Appendix: Table 2) showed a significant interaction between Mood and Time Stamp (β = -0.04, 95% C.I. = -0.08 –-0.00, p = .048) suggesting that in the Happy condition arousal scores decreased somewhat over time.

## Relationship between self-rating and FaceReader scores

Finally, we calculated Pearson's product-moment correlations to test the association between self-rated affective valence and arousal and the corresponding FaceReader scores, averaged across the duration of the AIP. To ensure reliable aggregate measurements FaceReader scores were averaged only for those ninety three participants with more than 1000 measurement points. There was no significant correlation (valence: r (91) = -.001, p = .99; arousal: r (91) = .0001, p = .99).

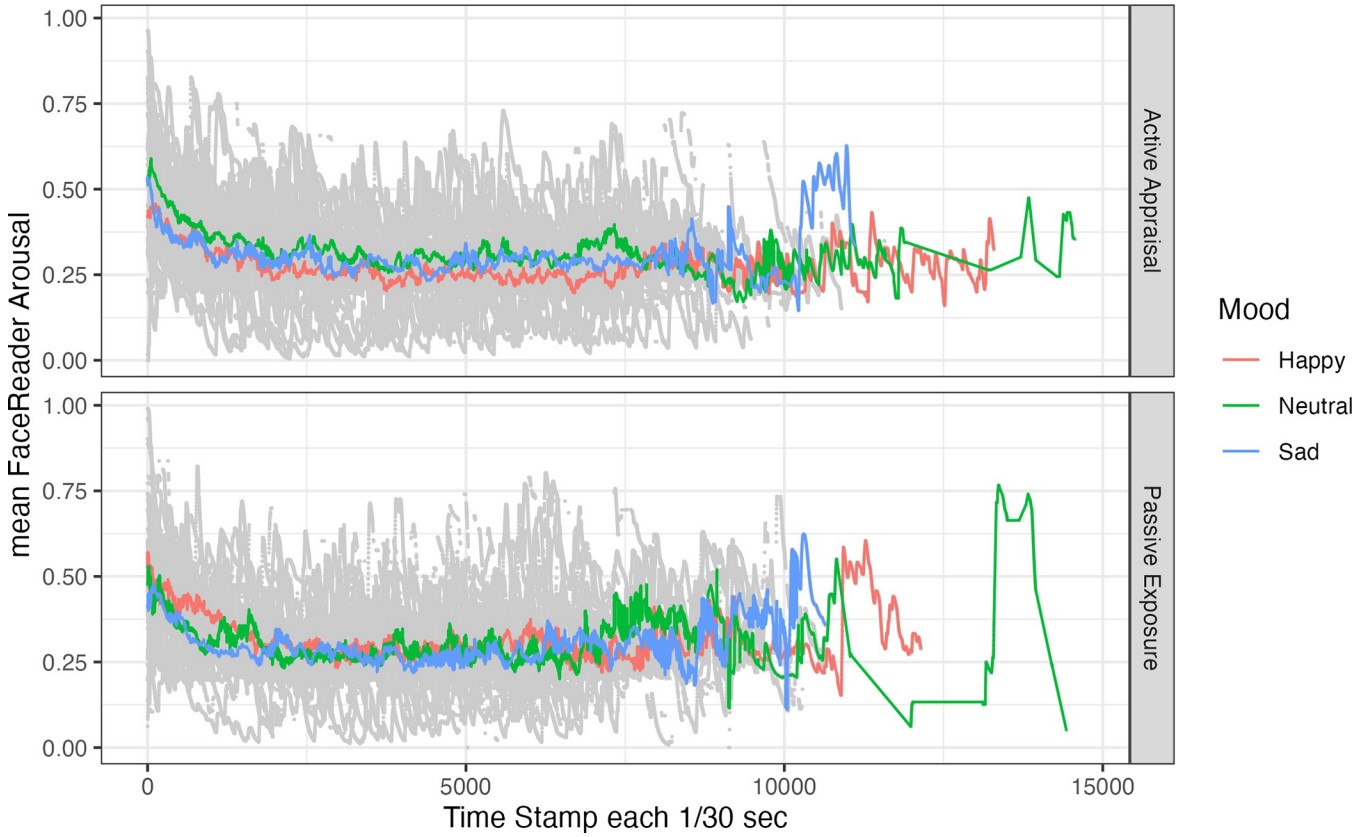

**Fig 4. Mean FaceReader arousal scores (in colour) and individual data points by participant (in grey) over time.** *Note.* Sessions differed in length resulting in wider fluctuations of the condition means for later time stamps with fewer data points.

## Discussion

We used an image-music AIP under two conditions—active appraisal and passive exposure—to examine whether participants' subjective valence ratings, provided after the AIP, are subject to response biases. We reasoned that if participants respond primarily based on demand characteristics and social expectations associated with the observed affective stimuli then there should be no difference in subjective ratings between the appraisal conditions. We found that participants' subjective valence ratings reflected the affective valence targeted by each AIP-condition: affective valence scores in the Sad condition were significantly lower, and affective valence scores in the Happy condition were significantly higher than those in the Neutral condition, as intended by the induction procedure. Crucially, the ratings were attenuated in the Active Appraisal compared to the Passive Exposure condition: A significant interaction indicated that the difference in subjective valence ratings between the Sad and Happy condition was smaller in the Active Appraisal condition. There were no effects of the AIP on subjective arousal ratings.

As discussed in the Introduction, there are several possible explanations for an effect of Appraisal condition on subjective valence ratings. One possibility is that participants genuinely felt less intense affect in the Active appraisal conditions, because affect was attenuated due to increased cognitive load caused by having to pay attention to where on the scale to click to perform an appraisal. However, it is also possible that participants' moods were no more intense in the Passive appraisal conditions, but that the extent to which demand characteristics and

social desirability influenced their self-reports was greater in this condition. If response biases were to account for all of the variance in participants' self-reported affective valence, however, we would expect the type of appraisal to make no difference at all. If participants were simply responding in the way they felt they ought to, having witnessed particular images, then those images should be all that matters, e.g. a picture of a starving child should elicit the same report of feigned sadness regardless of whether it has been appraised as such or not. While we cannot exclude that demand characteristics and social desirability play some role in participants' self-reports, our data provide evidence that they are not solely responsible for effects of AIPs and that relative differences between participants' self-reported moods represent meaningful individual differences in experienced affective valence. Even if the absolute scores of aggregate affective valence are biased to a certain extent—for example, the average valence rating of 36.68 for happy participants in the Active Appraisal condition could potentially also reflect the affective valence of happy participants in the Passive Exposure condition despite their stated average of 51.00, if the latter participants are more prone to exaggeration due to response biases—the relative difference in valence scores between groups appears to be robust.

Based on these findings, we argue that self-reports of affective states can be endorsed as informative measures of the affective states that participants experience. However, it should be borne in mind that self-reports may have other domain-specific limitations. Individuals differ in their degree of alexithymia, the inability to recognise one's own and others' emotions [65]. There can be substantial individual differences not only in participants' *willingness* to accurately report emotional states, but also in participants' awareness of them [16]. Even individuals low in alexithymia may be unaware of some unconscious emotional states [35]. Individuals also differ in their degree of *emotional granularity*—the precision with which they can specify their emotional states. As such, if a participant reports that they feel happy, this could refer to a very specific emotional state, or to a more generally pleasant level of emotional valence, depending on their emotional granularity [66]. It can therefore not be assumed that the happiness reported by one participant is the same as the happiness reported by another [35]. Likewise, when presented with emotional attributes that do not necessarily reflect their exact experience, participants may try to use the attributes they are given to report their own feelings (e.g. an excited participant may use the item "happy" to report their excitement [35]). Shorter scales in particular can suffer from measurement error, which can be reduced through aggregation across many items [67]. With sixteen items, the BMIS [60], measures affective valence by aggregation and therefore reduces error variance. However, the original version used only a four-point scale, which is not particularly sensitive to measuring subtle variability in one's mood. For the present study we therefore amended the BMIS by introducing a visual analogue scale, as well as new adjectives to expand the scope for aggregation. This enabled us to measure affective valence in a more sensitive and reliable manner to minimise the impact of the aforementioned limitations. Ultimately, our findings showed that when participants were asked to rate their affective states on a range of attributes the overall valence of their ratings is congruent with the affective valence targeted by the AIP.

We also tested whether facial expressions identified by FaceReader are more reliable indicators of mood targeted by AIPs. In contrast to the findings by Höfling et al. [26] and Noordewier and van Dijk [25], we did not find any effect of induced affective valence on FaceReader valence scores, and no effect of Appraisal conditions either. Crucially, these studies manipulated affective states in a within-participants design, whereas ours was a between-participants design. This suggests that FaceReader does not reliably detect variation in facial expressions associated with specific affective states across different individuals. In fact, as Fig 3 shows, there was considerable variability in FaceReader valence scores. Moreover, there was no correlation between subjective valence ratings and average FaceReader valence scores. Instead, we

found a small albeit significant decrease in arousal scores over time in the Happy condition. If this finding is genuine, it may suggest that participants exposed to pleasant images and happy music may have become less aroused over time. This finding does not match the results from participants' self-reports, and is difficult to account for. It needs further corroboration in a study that minimises variation in the duration of AIPs to rule out that this effect is carried mainly by the participants with longer sessions. Of course, given the discrepancy between self-reported and FaceReader-rated results, this finding could also be a Type I error. If it is replicable, it could indicate that our combination of happy images and music may induce not just a sense of happiness but also of calm and serenity.

The lack of a valence effect on FaceReader scores may have several reasons: First, FaceReader has been trained on posed photographs, and may not be able to recognise genuine dynamic expressions of emotion. Secondly, studies that showed AIP effects on FaceReader scores used a within-participants design but there has to date been no evidence that automated emotion recognition from facial expressions generalises across participants. In line with findings that facial expressions vary greatly across individuals [37, 38, 39] our study also suggests caution in using such automated systems for identifying emotions from faces across different individuals. This is true even if differences in hardware and in distance to camera may have introduced further noise into the data; after all, automated systems like FaceReader are advertised to be usable in a wide variety of not necessarily well-controlled contexts. Finally, participants watched the affective stimuli in solitude as the experimenter was hiding away from their view. In the absence of a social partner, the signalling function of behavioural affect indicators seems to be reduced leading to attenuated facial expressions [68, 69].

More generally, the failure to find reliable valence indicators in the automated analyses of facial expressions is in line with the Theory of Constructed Emotion [42, 70] which casts doubt on the possibility to identify 'objective' indicators that allow for reliable recognition of affective states across individuals and cultures. Given that different indicators of emotion frequently do *not* correlate, or only correlate weakly [16, 35, 71], and that there is no clear, objective way of distinguishing when a subject is experiencing a particular emotional state [35, 70, 72], it seems neither feasible nor necessary to seek objective verification of participants' self-reported affective states. Moreover, the search for objectivity in the measurement of emotion may commit a category mistake because emotion is not necessarily an objective phenomenon in the first place. If a person claims to be sad, then—even if observable behaviour and measurable physiology appear inconsistent with this claim–who else could ever be sufficiently qualified to correct the person? While the scientific study of emotion typically considers emotion as more than just the subjective feeling [46, 67], it is typically this feeling that defines an emotional episode. Fundamentally, what most would consider the defining feature of emotion, or specifically of emotional valence, is the subjective experiential, cognitive and/or reflective components, rather than the objective physiological or behavioural components [40, 49, 50, p. 29–30].

If one accepts the Theory of Constructed Emotion [42, 70], then an emotional experience such as sadness may be constructed in a multitude of different ways, with the variety of possible associated behavioural and physiological indicators being, for all practical purposes, virtually limitless. As a result, when self-reported affective states fail to correlate with 'objective' indicators, the self-report should not be automatically assumed to be the faulty measure. When the question itself is subjective, then subjective answers are *ipso facto* the most "objective" answers available. What we have shown here is evidence that such subjective reports at least partially reflect genuine subjective experiences rather than just demand characteristics and social expectations. Thus, our findings can be taken as evidence that multi-modal AIPs are efficacious inducers of target affective states, and that it is valid to verify these affective states via self-reports.

## Supporting information

**S1 Appendix. Modifications to the BMIS-VA during the experiment.**
(DOCX)

**S2 Appendix. Linear regression models for self-rated valence and arousal.**
(DOCX)

**S3 Appendix. Growth-curve analysis of FaceReader outputs.**
(DOCX)

## Acknowledgments

The authors wish to thank Konstantina Chatzispyrou, Valeria Ehlers, Valentina Sebastiani, Jordan Sculley, and Petra Susko for help with conducting the experiment.

## Author Contributions

**Conceptualization:** David J. Moulds, Jona Meyer, Janet F. McLean, Vera Kempe.

**Data curation:** David J. Moulds, Vera Kempe.

**Formal analysis:** David J. Moulds, Vera Kempe.

**Funding acquisition:** Vera Kempe.

**Investigation:** David J. Moulds, Jona Meyer, Janet F. McLean.

**Methodology:** Jona Meyer, Janet F. McLean, Vera Kempe.

**Project administration:** David J. Moulds, Vera Kempe.

**Supervision:** David J. Moulds, Janet F. McLean, Vera Kempe.

**Writing – original draft:** David J. Moulds, Janet F. McLean, Vera Kempe.

**Writing – review & editing:** David J. Moulds, Janet F. McLean, Vera Kempe.

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
