## [Decision Letter · Decision Letter 0]

13 Mar 2023

PONE-D-22-30135Exploring effects of response biases in affect induction proceduresPLOS ONE

Dear Dr. Kempe,

Thank you for submitting your manuscript to PLOS ONE. After careful consideration, we feel that it has merit but does not fully meet PLOS ONE’s publication criteria as it currently stands. Therefore, we invite you to submit a revised version of the manuscript that addresses the points raised during the review process.

We look forward to receiving your revised manuscript.

Kind regards,

Daniela Maria Romano

Academic Editor

PLOS ONE

“This study was supported by an RLINCS-studentship from Abertay University to the first author.”

Please state what role the funders took in the study.  If the funders had no role, please state: ""The funders had no role in study design, data collection and analysis, decision to publish, or preparation of the manuscript.

Additional Editor Comments:

Dear Authors

please attend to the suggestions of the reviewers. Highlight in yellow the changes made in the new manuscript, and submit the new version and a report on how the reviewers comments have been taken into consideration.

Reviewer 1

In the introduction, when building the argument of self-reports against behavioural or physiological measures (line 72-74), should have more references to back up the argument (recent publications).

When the authors state the need to investigate the biases in self-reports should have a paragraph explaining what has been done up to date in this area (line 74-77).

The same issue occurs in (line 84) when stating the need to use facial expressions. It should be an up-to-date paragraph stating the need to also include FACS in your investigation. Even though after proposing using FACS, you show examples, the build-up to find the need or gap should be further explained and explicit, especially if you then expose the positives and several negative reasons for using physiological indicators for this kind of research which is latter confirmed in the results.

The study is well presented, with relevant references that support the study design. There is a “NOTE” in line 195- 220 that explain different outcomes of possible results that are redundant and without references to back up the claims; maybe this should go in the discussion?

METHOD:

Four years to collect the data? please explain

Reviewer 2

The manuscript is clearly written and well-motivated. I think this study is a useful incremental contribution to the literature and ongoing discussion of the validity of AIPs. I would encourage the authors to consider discussing how demand effects may be moderated by experiment presentation (in other words, is it possible that active appraisal changes how participants view the goals or demands of the experiment?). However, I do not believe this request is sufficient to waylay publication.

Best Regards

Reviewers' comments:

Reviewer's Responses to Questions

**Comments to the Author**

1. Is the manuscript technically sound, and do the data support the conclusions?

Reviewer #1: Yes

Reviewer #2: Yes

2. Has the statistical analysis been performed appropriately and rigorously? 

Reviewer #1: Yes

Reviewer #2: Yes

3. Have the authors made all data underlying the findings in their manuscript fully available?

Reviewer #1: Yes

Reviewer #2: Yes

4. Is the manuscript presented in an intelligible fashion and written in standard English?

Reviewer #1: Yes

Reviewer #2: Yes

5. Review Comments to the Author

Reviewer #1: This paper explores the effects of affect-induced procedures such as self-reports or ratings (AIPs) in a between-participants design by using images and music to induce affect/mood by measuring actively and passively participants, at the same time, if they are correlated with facial expressions to understand if the responses are genuine or have cultural or context biases.

In the introduction, when building the argument of self-reports against behavioural or physiological measures (line 72-74), should have more references to back up the argument (recent publications).

When the authors state the need to investigate the biases in self-reports should have a paragraph explaining what has been done up to date in this area (line 74-77).

The same issue occurs in (line 84) when stating the need to use facial expressions. It should be an up-to-date paragraph stating the need to also include FACS in your investigation. Even though after proposing using FACS, you show examples, the build-up to find the need or gap should be further explained and explicit, especially if you then expose the positives and several negative reasons for using physiological indicators for this kind of research which is latter confirmed in the results.

The study is well presented, with relevant references that support the study design. There is a “NOTE” in line 195- 220 that explain different outcomes of possible results that are redundant and without references to back up the claims; maybe this should go in the discussion?

METHOD:

Four years to collect the data?

162 participants, In 6 conditions (ten males and ten females per condition)

The participants are well described, as well as the materials in the test and the protocols.

MATERIALS AND MEASURES:

The experiment is well presented, with several steps to reduce error, with appropriate ethics and randomised.

Results:

The statistical tests are suitable for this type of data, especially for linear regression.

Reviewer #2: The authors report an interesting experiment investigating the validity of affect induction procedures (AIP). Specifically, consenting participants underwent a traditional AIP (i.e., sad, happy, or neutral images + audio) and then completed a series of self-ratings. Crucially, in a between-participants design, half of the participants passively viewed the affective imagery, whereas the other half were required to actively appraise them. The logic of the experimental design is that, if self-ratings following AIP are subject to demand effects, then there should be no difference between the two experimental conditions (i.e., the demand effects are the same in both conditions). The author find a significant attenuation in affect-congruent self-ratings for the active appraisal condition compared to the passive viewing condition. Automatic coding of emotion via facial expressions was found to be a poor measure and unimpacted by AIP.

The authors have graciously made their data publicly available. Because of this, I was able to independently verify their statistical findings. The attenuation in mood-congruent ratings is significant despite the between-participants design and relatively small number of participants per bin. In fact, this interaction remains significant under a quite a number of robust statistics (e.g., robust regression using the Student-t distribution, laplace regression to evaluate differences in medians, etc.). As such, I am confident in their findings as reported.

The manuscript is clearly written and well-motivated. I think this study is a useful incremental contribution to the literature and ongoing discussion of the validity of AIPs. I would encourage the authors to consider discussing how demand effects may be moderated by experiment presentation (in other words, is it possible that active appraisal changes how participants view the goals or demands of the experiment?). However, I do not believe this request is sufficient to waylay publication.

In all, kudos to the authors for this interesting study and contribution to the literature!

6. PLOS authors have the option to publish the peer review history of their article (what does this mean?). If published, this will include your full peer review and any attached files.

Reviewer #1: **Yes: **Felipe Sheward

Reviewer #2: No

---

## [Author Response · Author response to Decision Letter 0]

3 Apr 2023

Dear Editor,

Thank you for processing our submission and sending the helpful comments from the reviewers. Below we explain how we have addressed their concerns and suggestions. We also have amended the reference list as requested by the reviewers.

Sincerely,

Vera Kempe (for all authors)

Reviewer 1

In the introduction, when building the argument of self-reports against behavioural or physiological measures (line 72-74), should have more references to back up the argument (recent publications).

Response: We agree with Reviewer 1 that it would be important to provide more recent evidence. However, an extensive search confirmed our assertion, expressed in the draft in line 68, that this issue has faded from attention in more recent work. To support this, we have now included further references in lines 71 and 74 which have also been cited in a recent meta-analysis (Joseph et al, 2020) which are all of an earlier date. 

When the authors state the need to investigate the biases in self-reports should have a paragraph explaining what has been done up to date in this area (line 74-77). 

Response: Despite occasional concerns about biases in self-reports in the literature, our main point is that not much has been done in this area since the compelling meta-analysis by Westermann et al. from 1996. This is precisely what we are trying to remedy with our work. Our aim is clarified by the following sentence in line 68 ‘More recently, these discussions seem to have faded from attention even though the issues are far from settled, and most manipulation checks of AIPs still use self-reports.’ We hope the reviewer can agree with our concern that this issue needed to be taken up and scrutinised more thoroughly. 

The same issue occurs in (line 84) when stating the need to use facial expressions. It should be an up-to-date paragraph stating the need to also include FACS in your investigation. Even though after proposing using FACS, you show examples, the build-up to find the need or gap should be further explained and explicit, especially if you then expose the positives and several negative reasons for using physiological indicators for this kind of research which is latter confirmed in the results.

Response: We thank the reviewer for pointing out the lack of clarity on our motivation for focussing on FaceReader. We have now added a sentence in line 89 to this effect, which reads ‘However, human coding of facial expressions using FACS is costly as it requires extensive certified training of the coders (12,13).‘ We feel that adding another paragraph discussing FACS would unduly lengthen the paper and divert focus but we are happy to take editorial guidance on this.

The study is well presented, with relevant references that support the study design. There is a “NOTE” in line 195- 220 that explain different outcomes of possible results that are redundant and without references to back up the claims; maybe this should go in the discussion?

Response: We thought it is important to justify in the Introduction why we are employing a two-tailed hypothesis; we fear it would create confusion for the reader to leave this important consideration for the Discussion. We agree that justification for the hypothesised outcomes would gain in persuasiveness if motivated by previous work so we included references to the literature on the link between emotions and cognitive load and to the measurement effects in the field of emotion research. 

METHOD:

Four years to collect the data? please explain

Response: We understand the reviewer’s concern about our data collection procedure but data collection had to be interrupted due to personal circumstances of one of the authors. 

Reviewer 2

The manuscript is clearly written and well-motivated. I think this study is a useful incremental contribution to the literature and ongoing discussion of the validity of AIPs. I would encourage the authors to consider discussing how demand effects may be moderated by experiment presentation (in other words, is it possible that active appraisal changes how participants view the goals or demands of the experiment?). However, I do not believe this request is sufficient to waylay publication.

Response: Wе thank Reviewer 2 for the positive evaluation of this work. We agree that it is possible that participants in the Passive Exposure condition are subject to demand characteristics in the sense that their mood reporting is affected by an appraisal of the images in addition to a reporting of their affective state. However, this would mean that in the Active Appraisal condition participants have already completed their appraisal when asked to report on their mood indicating the genuine effect of the mood induction above and beyond any appraisal-related demand characteristics. We have now clarified this in the Introduction in line 203-204.

---

## [Editor Report · Decision Letter 1]

2 May 2023

Exploring effects of response biases in affect induction procedures

PONE-D-22-30135R1

Dear Dr. Kempe,

We’re pleased to inform you that your manuscript has been judged scientifically suitable for publication and will be formally accepted for publication once it meets all outstanding technical requirements.

Within one week, you’ll receive an e-mail detailing the required amendments. When these have been addressed, you’ll receive a formal acceptance letter, and your manuscript will be scheduled for publication.

Kind regards,

Daniela M. Romano

Academic Editor

PLOS ONE

---

## [Editor Report · Acceptance letter]

4 May 2023

PONE-D-22-30135R1 

Exploring effects of response biases in affect induction procedures 

Dear Dr. Kempe:

I'm pleased to inform you that your manuscript has been deemed suitable for publication in PLOS ONE. Congratulations! Your manuscript is now with our production department. 

Kind regards, 

on behalf of

Professor Daniela Maria Romano 

Academic Editor

PLOS ONE